# Effects of Dietary Nano-Zinc Oxide Supplementation on Meat Quality, Antioxidant Capacity and Cecal Microbiota of Intrauterine Growth Retardation Finishing Pigs

**DOI:** 10.3390/foods12091885

**Published:** 2023-05-04

**Authors:** Shun Chen, Binbin Zhou, Jiaqi Zhang, Huijuan Liu, Longfei Ma, Tian Wang, Chao Wang

**Affiliations:** College of Animal Science and Technology, Nanjing Agricultural University, Nanjing 210095, China

**Keywords:** intrauterine growth retardation finishing pigs, nano-zinc oxide, meat quality, antioxidant capacity, intestinal microbiota

## Abstract

As nano-zinc oxide (Nano-ZnO), a new type of nanomaterial, has antioxidant and intestinal protection effects, we hypothesized that dietary Nano-ZnO could modulate poor meat quality, oxidative stress and disturbed gut microbiota in the finishing pig model of naturally occurring intrauterine growth retardation (IUGR). A total of 6 normal-born weight (NBW) and 12 IUGR piglets were selected based on birth weight. The pigs in the NBW group received a basal diet, and IUGR pigs were randomly divided into two groups and treated with basal diet and 600 mg/kg Nano-ZnO-supplemented diet. Dietary Nano-ZnO ameliorated IUGR-associated declined meat quality by lowering the drip loss_48h_, cooking loss, shearing force and *MyHc IIx* mRNA expression, and raising the redness (*a**), peak area ratio of immobilized water (*P*_22_), sarcomere length and *MyHc Ia* mRNA expression. Nano-ZnO activated the nuclear factor erythroid 2-related factor 2-glutamyl cysteine ligase (Nrf2-GCL) signaling pathway by promoting the nuclear translocation of Nrf2, increasing the GCL activities, and mRNA and protein expression of its catalytic/modify subunit (GCLC/GCLM), thereby attenuating the IUGR-associated muscle oxidative injury. Additionally, the composition of IUGR pigs’ cecal microbiota was altered by Nano-ZnO, as seen by changes in Shannon and Simpson indexes, the enhanced *UCG-005*, *hoa5-07d05 gut group* and *Rikenellaceae RC9 gut group* abundance. The *UCG-005* and *hoa5-07d05 gut group* abundance were correlated with indicators that reflected the meat quality traits and antioxidant properties. In conclusion, Nano-ZnO improved the IUGR-impaired meat quality by altering water holding capacity, water distribution and the ultrastructure of muscle, activating the Nrf2-GCL signaling pathway to alleviate oxidative status and regulating the cecal microbial composition.

## 1. Introduction

Intrauterine growth retardation (IUGR) usually refers to multiple internal and external stimuli to the mammalian mother during gestation, resulting in retardation of fetal growth in the mother’s uterus and failure to achieve genetic growth potential, as evidenced by low initial birth weight and impaired tissue and organ development [1,2,3]. Oxidative stress was continuously found in critical tissues (skeletal muscle, islets and liver) in nascent and adult individuals exposed to IUGR [4]. Given its high unsaturated fatty acids and protein content, porcine meat is exceptionally susceptible to oxidation, which causes a decrease in meat quality and even spoilage [5,6]. Past research has shown that the negative impact of IUGR on meat quality results from the elevated reactive oxygen species level and the lowered antioxidant capacity [7,8]. The skeletal muscle fiber is a dynamic structure, and the alteration of the percentage of muscle fiber types and the decrease in sarcomere length are closely correlated to the occurrence of oxidative stress in the muscles [9,10]. Moreover, the disordered antioxidant enzyme activity and accumulation of peroxide products adversely affects the molecular structure and contractile function of muscle fibers. Consequently, the decline of meat quality in IUGR pigs may partly result from the changed antioxidant status, fiber types and sarcomere length of muscle.

As an inseparable “organ” of the body, the intestinal microbiota greatly influences the host’s growth and development, immune inflammation, biological barrier function and other physiological functions [11]. It was well documented that IUGR strongly shapes the gut microbiota composition, contributes to dysbiosis of the gut microbiota and raises the risk of metabolic disease [12,13]. An increasing number of studies have also concentrated on the relationships between skeletal muscle metabolism and intestinal microbiota in recent years [9,14]. It has been reported that the increased *Lactic acid bacteria* and decreased *Enterobacteriaceae* levels are related to pork’s water-holding capacity (WHC) and color traits [15]. The *WPS-2* abundance in the colon of finishing pigs was shown to be positively correlated to its muscle lactate contents and drip loss but negatively correlated to pH_45min_ [9,16,17]. Consequently, it is logical to hypothesize that the decreased meat quality in IUGR pigs may correlate with the alterations of intestinal microbiota.

As industry increasingly applies nanotechnology to the feed additive industry, the prospect of nano-zinc oxide (Nano-ZnO) in improving meat quality and antioxidant capacity and regulating intestinal microbiota of animals has attracted the attention of experts at home and abroad [18,19,20,21,22]. Our previous work also demonstrated that Nano-ZnO supplementation showed favorable effects on improving the oxidative status of jejunal mucosa in pigs suffering from IUGR via the activation of nuclear factor E2-related factor 2 (Nrf2) pathway [23]. Nevertheless, it remains unclear whether the intestinal microbiota plays a role in Nano-ZnO-mediated attenuation of the inferior meat quality and decreased antioxidant capacity of IUGR pigs. Therefore, the aim of this study was to explore the impact of Nano-ZnO on alleviating the lowered meat quality, diminished antioxidant capability and imbalanced cecal microbiota structure of IUGR pigs.

## 2. Materials and Methods

### 2.1. Experimental Design, Diets, and Management

The Nanjing Agricultural University Institutional Animal Care and Use Committee authorized and supervised all experiments (Nanjing, China. Permit number SYXK-2019-00142).

In the study, six sows in good health, with comparable parity and litter size, were selected. From each litter, one NBW male piglet (1.52 ± 0.01 kg) and two IUGR male piglets (0.96 ± 0.02 kg) [Duroc × (Landrace × Yorkshire)] were chosen. Normal birth weight (NBW) refers to newborn piglets who weigh within 0.5 standard deviations (SD) of their littermates’ mean birth weights (BW), while IUGR refers to newborns whose BW is 2 SD lower [24]. Then, the selected 6 normal-born weight piglets were allocated to the NBW group, while the 12 piglets suffering from IUGR were divided into the IUGR and IUGR + Zn groups (each group comprised 6 pens, each pen comprised 1 piglet) at random. At the age of 21 days, all piglets were weaned. Following weaning, the pigs in the NBW and IUGR groups received a basal diet, while the IUGR + Zn group received a basal diet supplemented with 600 mg/kg Nano-ZnO (95% purity). Before this investigation, we conducted an independent experiment on weaned piglets to select the best suitable dosage [25], and we therefore speculated that 600 mg/kg Nano-ZnO would be the ideal dose for long-term addition. Identical to the Nano-ZnO utilized in the prior study [25], the Nano-ZnO was purchased from Zhangjiagang Bonded Area Hualu Nanometer Material Co., Ltd. in Nanjing, China. All the piglets had unrestricted access to feed and water and were fed in a regulated atmosphere (room temperature of 28 ± 2 °C and 22 ± 2 °C in the nursery period and fattening period, respectively, relative humidity of 55 ± 5%) throughout the experiment. The basal diets were formulated in accordance with the National Research Council (2012) guidelines [26], and their components and nutrient content are presented in Appendix A. 

### 2.2. Sample Collection and Preparation

All of the finishing pigs were sent to the slaughter room on day 163, where they were allowed to drink freely but not to feed for 12 h before slaughter. After that, pigs from each group were electrocuted, bled, scalded, dehaired, skinned and disemboweled in accordance with regular commercial practice. Approximately 10 g of *longissimus dorsi* (LD) samples between the 9th and 11th ribs were obtained within 20 min of slaughter, rapidly frozen with liquid nitrogen for analysis of mRNA expression, Western blotting, and enzyme activity. A portion of muscle tissue from the same left LD region was cut into 1 μm slices and promptly stored in 2.5% glutaraldehyde solution for transmission electron microscope (TEM) investigation. About 50 g LD samples were separated between the 11th and 13th ribs for subsequent analysis. In addition, the cecum digesta was collected and kept in sterile freezing tubes at −80 °C for further investigation.

### 2.3. Meat Quality Analysis

The pH levels of LD were assessed by a portable pH meter (pH-STAR, Mattuas, Hamburg, Germany) at 45 min (pH_45min_) and 24 h (pH_24h_) after slaughter. The meat color (luminance, *L**; redness, *a**; yellowness, *b**) was determined by the Minolta chroma meter (CR-10; KonicaMinolta Sensing, Inc., Osaka, Japan) at 24 h after slaughter. The colorimeter was calibrated using white tiles in line with the operating instructions before use. For the indicators of pH and meat color, the mean value was computed after measuring at least three points on each piece of the sample.

Additionally, after storing for 24 and 48 h, the drip loss data were determined. Briefly, 10 g of muscle fragments from the LD tissues were taken, weighed as the beginning weights (M_1_), kept in plastic bags and at 4 °C. After 24 h, the samples were removed, dried with filter paper, and weighed to estimate M_2_. After 48 h, the same procedures were repeated to obtain M_3_. The drip loss (%) at 24 and 48 h were calculated using the following equations: drip loss_24h_ (%) = [(M_1_ − M_2_)/M_1_ × 100%]; drip loss_48h_ (%) = [(M_1_ − M_3_)/M_1_ × 100%].

The method used to calculate the cooking loss was modified from that given in our previous research [27]. Concisely, each sample of about 15 g from LD was separated into plastic bags and weighed (M_before_). The samples were then cooked for 10 min in 80 °C water until they achieved an internal temperature of 75 °C. Subsequently, the samples were left to cool to room temperature under running water and then reweighed (M_after_) with filter paper to absorb the water from the surface. The following equation was used to compute the cooking loss (%): cooking loss (%) = [(M_before_ − M_after_)/M_before_ × 100%].

The shearing force was determined by Digital Meat Tenderness Meter (C-LM3B, Northeast Agricultural University, Harbin, China) following the method used in the earlier investigation, with minor adjustments [28]. Three subsamples of each sample were recorded; then, the mean values were determined.

### 2.4. Low-Field NMR Transverse Relaxation (T_2_) Analysis

The peak area ratio (*P*_21_, *P*_22_ and *P*_23_) and the relaxation time (*T*_21_, *T*_22_ and *T*_23_) of bound water, immobilized water and free water of LD were determined by PQ001 LF-NMR analyzer (Shanghai Niumag Electronic Technology Co., Ltd., Shanghai, China). The magnet’s temperature was set at 32 °C, the spectrometer’s frequency was set at 0.5, and the magnetic field was set at 22.4 MHz. Briefly, a sample of about 2 g of the LD was taken, cut and placed into an NMR tube, then the tubes were put in the LF-NMR magnet cavity to obtain *T*_2_ relaxation characteristics, which were then evaluated using the Carr–Purcell–Meiboom–Gill (CPMG) sequence. The following parameters were utilized in the data gathering process: repetition time (TR) is 4500 ms, half echo time (τ) is 200 μs, echo count (EC) is 3000, number of repeated scans (NS) is 8 and π-value is 110 μs. 

### 2.5. Ultrastructural Analysis

The LD samples that had been fixed in 2.5% glutaraldehyde solution for 24 h at 4 °C were then fixed in 1% osmium for 1 h. Then, the samples were dehydrated using a succession of gradient ethanol solution and treated by, in sequence, pure acetone, a mixture of pure acetone and embedding agent, and pure embedding agent. Finally, the microtome (Boeckeler, Tucson, AZ, USA) was used to cut the ultrathin sections, and the TEM (Hitachi H-7650, Tokyo, Japan) was used to observe the photos. Following the method used in the earlier investigation [10], the sarcomere lengths of LD were examined.

### 2.6. Real-Time qPCR Analysis

To evaluate the relative mRNA expression in LD, the previous methods [25,29] were utilized to extract the total RNA, reverse-transcribe the RNA into cDNA and amplify the cDNA by quantitative real-time PCR system. The sequences of specific primers for the target and housekeeping genes are shown in Appendix A.

### 2.7. Determination of Antioxidant and Glutathione-Related Indicators

The contents of malondialdehyde (MDA, No. A003-1-2), protein carbonyl (PC, No. A087-1-1), glutathione (GSH, A061-1-2) and oxidized glutathione (GSSG, A061-1-2), the levels of total antioxidant capacity (T-AOC, No. A015-2-1), and the activities of superoxide dismutase (SOD, No. A001-3-2), catalase (CAT, No. A007-2-1), glutathione peroxidase (GSH-Px, No. A005-1-2), glutathione reductase (GR, No. A062-1-1) and glutathione S-transferase (GST, No. A004-1-1) were measured using commercial kits (Nanjing Jiancheng Bioengineering Institute (Nanjing, China)). The activities of γ-glutamyl cysteine ligase (γ-GCL, No. BC1210) were measured by commercial kits provided by Beijing Solarbio Science & Technology Co., Ltd. (Beijing, China). All procedures were carried out exactly as specified by the manufacturers.

### 2.8. Western Blotting

In accordance with the prior method [8], the Western blotting analysis was carried out with the LD tissues. Briefly, proteins of various sizes were separated using SDS-PAGE after being extracted, quantified and denaturalized. Then, the target protein was transferred to the PVDF membranes. After incubating with primary and secondary antibodies, the membranes were blocked in 5% skimmed milk. To analyze the nuclear Nrf2 protein expression, the nuclear protein was obtained by commercial kits (Beijing Solarbio Science & Technology Co., Ltd., Beijing, China). The target protein expression was detected by the ChemiDoc MP Imaging System (Bio-Rad Laboratories, Inc., Hercules, CA, USA). Internal references for total and nuclear proteins were GAPDH and Lamin B1, respectively.

### 2.9. 16S rRNA Analysis of Cecal Microbiota

Following the manufacturer’s instructions, the total DNA was extracted from each cecum digesta sample (*n* = 6 for each group) using the E.Z.N.A.^®^ Soil DNA Kit from Omega Bio-tek, (Norcross, GA, U.S.). Using the AxyPrep DNA Gel Extraction Kit (Axygen Biosciences, Union City, CA, U.S.), amplicons were extracted from 2% agarose gels and purified according to manufacturers’ protocols.

The quantification of the purified PCR products was performed using Qubit^®^ 3.0 (Life Invitrogen, Carlsbad, CA, USA), and each of the 24 amplicons was combined evenly. According to Illumina’s method for preparing a genomic DNA library, the pooled DNA product was utilized to build an Illumina Pair-End library. After that, the Illumina MiSeq platform performed paired-end sequencing (2 × 250) of the amplicon library. The processed sequences were submitted to the DADA2 method to detect indel-mutations and replacements [30]. The Deblur denoising method was used to cluster the sequences into operational taxonomic units (OTUs) at 100% similarities [31].

In the current study, the species accumulation curve was used to determine the species richness and adequacy of sample size. The alpha diversity was analyzed using *Chao*, *Shannon* and *Simpson* indexes, while the beta diversity was analyzed using principal component analysis (PCoA) of unweighted UniFrac distances. The composition of the three groups of microorganisms is shown as the percentage of community abundance at phylum and genus levels. The LEfSe was used to determine the distinct microbiota species in each group by linear discriminant analysis (LDA). The Kruskal–Wallis test and Wilcoxon rank sum test were utilized to determine the differences between the three groups and the two groups.

### 2.10. Statistical Analysis

Except for the data on cecal microbiota, all statistics were analyzed by SPSS 26.0 (SPSS Inc., Chicago, IL, USA) software, and differences were assessed using a one-way ANOVA model and Duncan’s multiple range test. The statistical significance was set at *p*-value 0.05, and the data are presented as mean values with pooled standard errors of means (SEM). Moreover, the Pearson correlation coefficient was employed in correlation analysis (*n* = 6).

## 3. Results

### 3.1. Meat Quality

As indicated in Table 1, in comparison to the NBW group, IUGR exposure significantly increased the drip loss_48h_ in the LD tissues of pigs (*p* < 0.05). Additionally, dietary Nano-ZnO significantly decreased the drip loss_48h_, *a**, cooking loss and shearing force of IUGR pigs (*p* < 0.05). There was no significant difference in pH_45min_, pH_24h_, drip loss_24h_, *L** or *b** among the three groups (*p* > 0.05).

### 3.2. LF-NMR Results

As indicated in Table 2 and Figure 1, compared to the NBW group, IUGR pigs had higher *P*_21_ and lower *P*_22_ peak area ratios (*p* < 0.05). Moreover, compared to the IUGR pigs, dietary Nano-ZnO could significantly decrease the *P*_21_ peak area ratio (*p* < 0.05). No significance was found in the *P*_23_ peak area ratio among the three groups (*p* > 0.05).

### 3.3. Ultrastructure and Muscle Fiber Characteristics Analysis

The effects of dietary Nano-ZnO on the ultrastructure of LD tissues in IUGR pigs are illustrated in Figure 2A,B with representative photographs that demonstrate the results of sarcomere length statistics. Compared with the NBW group, pigs suffering from IUGR showed shorter (*p* < 0.05) and more irregular sarcomeres. The addition of Nano-ZnO to IUGR pigs significantly increased sarcomere length (*p* > 0.05) and improved myofibril structure. Furthermore, *MyHc IIx* mRNA expression was significantly decreased in the IUGR group compared with the NBW group (*p* < 0.05). Dietary Nano-ZnO significantly increased the *MyHc Ia* and *MyHc IIx* mRNA expressions (*p* < 0.05). The *MyHc IIa* and *MyHc IIb* mRNA expressions showed no significance among the groups (*p* > 0.05).

### 3.4. Evaluation of Oxidative Status

As indicated in Figure 3, the effect of Nano-ZnO on the oxidative status of LD tissues in the IUGR pigs was evaluated. The contents of MDA and PC were significantly increased (*p* < 0.05), and GSH-Px activities and T-AOC levels were significantly decreased (*p* < 0.05) in IUGR pigs compared to the NBW pigs. Following Nano-ZnO supplementation, the MDA and PC contents were significantly decreased (*p* < 0.05), and the activities of CAT and GSH-Px and the levels of T-AOC were significantly increased (*p* < 0.05). There was no significant change in SOD activities across groups (*p* > 0.05).

### 3.5. Glutathione-Related Indicators

In Figure 4, in comparison to the NBW pigs, IUGR led to lower contents of GSH and activities of GCL in the LD tissues of pigs (*p* < 0.05). However, Nano-ZnO supplementation significantly increased the contents of GSH, GSH: GSSG ratio, and the activities of GCL and GR, and decreased the contents of GSSG in the LD tissues of IUGR pigs (*p* < 0.05). No significant difference in the activities of GST among the groups was found (*p* > 0.05).

### 3.6. Genes Related to the Nrf2-GCL Signaling Pathway

Figure 5 demonstrates the mRNA expression levels of genes involved in the Nrf2-GCL signaling pathway. The *Keap1* mRNA expression was significantly increased, while the *GPX1* and *GR* mRNA expressions was significantly decreased in IUGR pigs compared to the NBW pigs (*p* < 0.05). Additionally, Nano-ZnO treatment significantly increased the *Nrf2*, *GCLM*, *GCLC*, *HO-1*, *GPX1*, *GR*, *GSTT1* and *GSTK1* mRNA expression levels, and decreased the *Keap1* mRNA expression in IUGR pigs (*p* < 0.05). The *GPX4* and *GSTT1* mRNA expression did not differ among the groups (*p* > 0.05).

### 3.7. Protein Expression

The protein expression of Nrf2 was significantly down-regulated and that of Keap1 was significantly up-regulated when the pig suffered from IUGR in comparison to NBW pigs (Figure 6, *p* < 0.05). While compared to IUGR pigs, IUGR pigs from Nano-ZnO treatment increased protein expression of nuclear Nrf2, total Nrf2, GCLM and GCLC, and decreased the protein expression of Keap1 (*p* < 0.05).

### 3.8. Gut Microbiota Analysis

As indicated in Figure 7A, its species accumulation curves leveled off when the number of samples reached 18, demonstrating that the sequencing data reached saturation and could cover the majority of cecal microbiota. Pigs suffering from IUGR showed lower Shannon index and higher Simpson index, while supplementation with Nano-ZnO effectively reversed the changes in both indexes (*p* < 0.05), suggesting that Nano-ZnO could improve the alpha diversity of cecum microbiota in pigs exposed to IUGR by increasing the Simpson index and decreasing the Shannon index (Figure 7B–D). Furthermore, the analysis of PCoA and similarities (Figure 7E) indicated a distinct separation among the structure of cecum microbiota (R = 0.4039, *p* = 0.026), and the PC1 and PC2 contributed to 11.66% and 9.31% of the overall variation, respectively. However, the microbiota community between IUGR and IUGR + Zn groups showed no remarkable difference (R = 0.1370, *p* = 0.143).

As shown in Figure 8A, *Firmicutes*, *Bacteroidota*, *Spirochaetota*, *Proteobacteria* and other phyla were found to make up the majority of the cecum microbiota, according to the taxonomic investigation at the phylum level. Figure 8B shows the relative abundance of cecal microbiota in the top 25 at the genus level. Relative abundance of cecal microbiota at the phylum level showed no significance among groups (Figure 8C). The relative abundance of *UCG-005*, *Rikenellaceae RC9 gut group* and *hoa5-07d05 gut group* in IUGR pigs was significantly decreased in comparison to the NBW pigs, while dietary Nano-ZnO increased the relative abundance of these three kinds of cecum microbiota significantly (Figure 8D, *p* < 0.05). The LDA analysis further revealed the enriched taxa on the OTU level in cecal microbiota among groups (Figure 8E). Dietary Nano-ZnO significantly increased the abundance of *Rikenellaceae_RC9_gut_group*, *Christensenellaceae*, *Christensenellales*, *Christensenellaceae_R_7_group*, *UCG_003*, *Lachnospiraceae_UCG_001*, *Sphingomonas*, *Eggerthellaceae*, *Amnipila* and *Incertae_Sedis* (*p* < 0.05).

### 3.9. Correlation Analysis

As indicated in Figure 9, the correlation between cecal microbiota and indicators reflecting meat quality and antioxidant properties showed that abundance of *UCG-005* is positively correlated with the GR activities, the GSH/GSSG ratio and *P*_22_, and negatively correlated with the GSSG and PC contents, and Shearing force. Moreover, the *hoa5-07d05 gut group* abundance is positively correlated with the CAT activities.

## 4. Discussion

Pigs with IUGR often exhibit deficiencies in the antioxidant system, resulting in declined meat quality [27]. The purpose of our study was to investigate the effect of Nano-ZnO supplementation on the meat quality of IUGR pigs and detect their underlying mechanisms. The pH, color, WHC and tenderness of meat are considered to reflect the freshness and health of meat and are highly valued by enterprises and consumers [16,17,32]. The current study demonstrated that IUGR pigs had higher drip loss at 48 h, whereas dietary Nano-ZnO relieved the negative impacts of IUGR on meat quality by decreasing the drip loss at 48 h, cooking loss and shearing force, and increasing the *a** value. Our results were consistent with earlier studies that found inferior meat quality in pigs exposed to IUGR [8,33]. Limited research demonstrated that broilers received zinc-bearing palygorskite showed decreased drip loss in the pectoral and thigh muscles [34], and finishing pigs receiving ZnO showed ameliorated meat quality through effects on the indicators of meat color characteristics and cooking loss [35]. The reduction of cooking loss and drip loss in our study indicated the superior effects of Nano-ZnO on improving the WHC of meat. To further evaluate the WHC of meat, a fast and nondestructive spectral detection technology, LF-NMR, was also applied to detect the water distribution of meat. The current study indicated that the peak area ratio of immobilized water (*P*_22_) contents was decreased in the LD tissues of the IUGR pigs, and after receiving Nano-ZnO, the *P*_22_ contents were increased, which suggested that Nano-ZnO supplementation could avoid the excessive conversion from the immobilized water to free water. The findings were parallel to the results of cooking loss and drip loss, suggesting that dietary Nano-ZnO alleviated the decreased WHC, and this may be explained partly by the increased *P*_22_ contents.

Furthermore, the muscle ultrastructure also affects meat quality traits, such as the sarcomere length affecting the tenderness and the WHC of muscle, with shorter sarcomere length often accompanying higher shearing force and lowered WHC [36,37]. Iruretagoyena et al. demonstrated that IUGR fetuses showed considerably lower sarcomere length compared to adequate growth for gestational age fetuses [38]. The present study also corroborated the decline in the sarcomere length in LD tissues of IUGR pigs and demonstrated that Nano-ZnO supplementation had favorable effects on attenuating the IUGR-associated decreased sarcomere length. This may help to partially explain why Nano-ZnO reduced the shearing force and increased the WHC in LD tissues of IUGR pigs. On the other hand, according to the contraction characteristics of muscles and the diversity of myosin heavy chain, the muscle fibers can be divided into four types (MyHc Ia, IIa, IIx and IIb isoforms) [39]. The present study demonstrated that the *MyHc IIx* mRNA expression was higher in the IUGR pigs, and Nano-ZnO supplementation increased the *MyHc Ia* mRNA expression and decreased the mRNA expression of *MyHc IIx*. Similarly, Felicioni et al. [40] revealed that IUGR pigs showed a lower percentage of MyHc Ia at 150 days of age. The research conducted by Paulk et al. found that feeding ZnO linearly decreased the type IIx fibers abundance in the longissimus lumborum muscles of pigs [36]. Moreover, it has been established that the myoglobin, the main protein that affects the redness of the meat, is more abundant in oxidized muscle fibers (MyHc Ia and IIa), which is closely related to meat quality traits [8,41,42]. Therefore, we inferred that the improved redness of meat in the IUGR + Zn group might be attributed to the modulatory effect of Nano-ZnO on muscle fiber composition.

Another primary element that alters the nutritional value and sensory perception of meat is its diminished antioxidant capabilities, and Nano-ZnO can regulate the levels of the main free radical scavengers (SOD, GSH-Px and CAT) of the animal antioxidant system, thus protecting the animal organism from damage caused by oxidative stress [23,43]. The current study’s findings support this viewpoint even more, showing that the LD tissues of IUGR pigs with poor meat quality also had higher lipid and protein peroxidation products (MDA and PC) contents and lowered GSH-Px and T-AOC levels, while Nano-ZnO supplementation substantially reversed these trends. The alteration of these indicators suggested that Nano-ZnO effectively ameliorated the decreased antioxidant status in the LD tissues of IUGR pigs via the enzymic and non-enzymic antioxidant systems.

Moreover, this study also demonstrated the powerful capability of Nano-ZnO in the Nrf2-GCL pathway activation, which follows a prior study that revealed that zinc treatment induced an increase in GCL-mediated GSH synthesis via the antioxidant response elements (ARE)-Nrf2 pathway [44]. Under a physiological environment, Keap1 binds to Nrf2, and Nrf2 activity is inhibited. However, as a result of oxidative stress, Nrf2 and Keap1 dissociate and translocate to the nuclear where they are activated and subsequently bind to the ARE, activating the downstream antioxidant factors GCLC, GCLM, GST and phase II enzymes and up-regulating GSH levels, thereby regulating the antioxidant defense system in reaction to oxidative stress. Furthermore, GSH-Px catalyzes the process to convert GSH to GSSG, while GR converts GSSG back to GSH, and the ratio of GSH: GSSG can be regarded as an essential indicator of the redox status in tissue cells [45,46,47]. Our results demonstrated that Nano-ZnO supplementation increased the ratio of GSH: GSSG, and GSH, GR and GCL activities, and reduced the GSSG contents in IUGR pigs, indicating that the modulation of GSH-related indicators may be a vital mechanism by which Nano-ZnO prevents the decline of antioxidant status in LD tissues of IUGR pigs. Our conjecture was further supported by the increased mRNA and protein levels of GCLC and GCLM and the mRNA expression of several necessary GSH metabolism-related catalytic enzymes in the IUGR + Zn group. Moreover, Nano-ZnO was also found to facilitate the transfer of Nrf2 into nuclear to exert antioxidant activity manifested by increased the expression of nuclear and total Nrf2 protein and decreased Keap1 protein expression in the IUGR + Zn group in the current study. The *Nrf2* and *Keap1* mRNA expression also showed a comparable trend. The above results provided more evidence for the crucial protective function of the Nrf2-GCL signaling pathway in the antioxidant effect of Nano-ZnO.

The existence of the muscle–gut microbiota axis has been validated by multiple investigations, implying a substantial link between the development and metabolism of muscle and gut microbiota [9,48,49]. As a result, our investigation analyzed the cecal microbiota composition via 16S rRNA gene sequencing. Indicators of community diversity, such as the Shannon and Simpson indexes, are frequently utilized [50]. The current study indicated that Nano-ZnO supplementation increased and decreased the Shannon and Simpson indexes of IUGR pigs, respectively. This finding suggests that the alpha diversity of cecal microbiota in IUGR pigs was considerably decreased, and Nano-ZnO showed a powerful capacity to regulate the dysbiosis of cecal microbiota by modulating the indexes reflecting the alpha diversity of cecal microbiota. Similarly, Pieper et al. found that adding 1000 and 2500 mg/kg of zinc oxide to weaned pigs raised the Shannon index and species richness in the ileum [51]. Although it has been demonstrated that the beta diversity of the microbiota community in rats that received Nano-ZnO showed distinct separated cluster compared with the control group [20], Nano-ZnO supplementation did not change the beta diversity of microbial community in IUGR pigs in the present study. The used dose, object and duration of nano-zinc oxide may all be responsible for this difference.

To further determine whether Nano-ZnO affects meat quality in IUGR pigs via regulating gut microbiota composition, we screened the taxa that differed most among treatments at the phylum and genus levels and conducted a correlation analysis between the cecal microbiota and indicators reflecting meat quality traits and antioxidant properties. In line with He et al. [52]., we observed that the *Firmicutes*, *Bacteroidaota* and *Spirochaetota* are the prevalent species at the phylum level, but the relative abundance of the top 10 phyla species showed no marked difference among the groups in the present study. Furthermore, *UCG-005* is identified as significantly different taxa at the genus level in the current study. Previous research has identified *UCG-005* as a potentially beneficial bacterium, and some antioxidants, such as chlorogenic acid, have been demonstrated to boost *UCG-005* abundance while increasing the antioxidant status of broilers [53]. The results of the rank sum test indicate that pigs suffering from IUGR had lower abundance of *UCG-005*. This result is comparable to that of Gaukroger et al., who found that low birthweight pigs had declined *UCG-005* abundance [54]. Nano-ZnO supplementation showed an effective capacity to increase *UCG-005* abundance, which is favorably associated with the GR activities and GSH/GSSG ratio and negatively correlated with the *T*_22_ value and PC contents. According to Liu et al., the *UCG-005* abundance is inversely correlated with finishing pigs’ longissimus dorsi muscle shearing force [55]. Furthermore, the current study also indicated that Nano-ZnO supplementation enriched the *hoa5-07d05 gut group* and *Rikenellaceae_RC9_gut_group* abundance, which both belong to the *Rikenellaceae* group [56]. The *hoa5-07d05 gut group* abundance was positively correlated with CAT activity. Although several studies reported a positive relationship between *Rikenellaceae* abundance and the fermentation of dietary fiber [57,58], it is yet unclear if the presence of the *Rikenellaceae* family has a direct impact on the muscle’s oxidative state. In summary, we tentatively proved that the enhanced antioxidant status and meat quality of LD tissues by Nano-ZnO supplementation could be partially explained by its modulating effects on the abundance of *UCG-005* and *hoa5-07d05 gut group* in the cecal microbiota, according to the correlation analysis. The present investigation may provide a new perspective on studying the regulatory effect of Nano-ZnO-mediated gut microbiota on the development and metabolism of muscle. However, the present study still has its limitations. The connection between these two taxa or their metabolites and host muscle metabolism still need to be sufficiently established. Therefore, a future study should conduct a fecal microbiota transplantation experiment should be carried out to determine if the microbiota can, in turn, regulate the development and metabolism of the host’s muscles.

## 5. Conclusions

In conclusion, Nano-ZnO supplementation could alleviate the IUGR-associated decline in meat quality by regulating the color, tenderness, drip loss, water distribution and ultrastructure of muscle. Moreover, Nano-ZnO showed the superior potency of ameliorating the decreased antioxidant capacity of LD tissues in IUGR pigs and activating the Nrf2-GCL signaling pathway. Nano-ZnO supplementation changed the alpha diversity and increased the *UCG-005* and *hoa5-07d05 gut group* abundance of cecal microbiota, which may closely correlate with the improved meat quality and enhanced antioxidant capacity.

## Figures and Tables

**Figure 1 foods-12-01885-f001:**
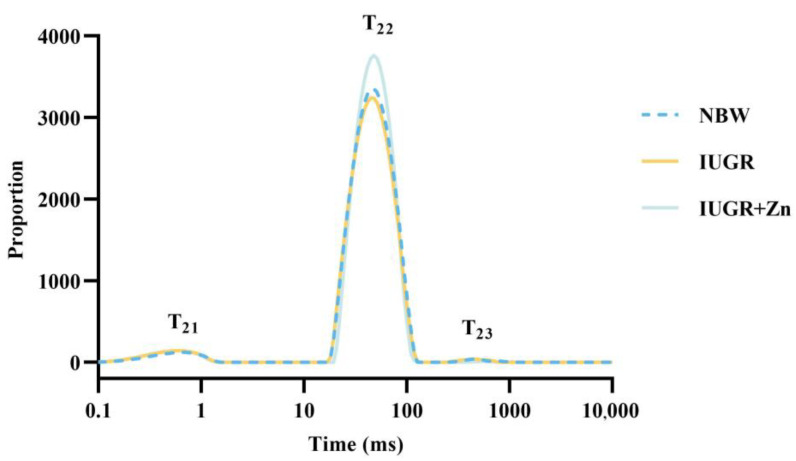
Effects of Nano-ZnO supplementation on the relaxometry time of LF-NMR of LD tissues in the IUGR pigs (*n* = 6). NBW, normal-born weight pigs; IUGR, intrauterine growth restriction pigs; IUGR + Zn, IUGR pigs fed with diets supplemented with 600 mg/kg Nano-ZnO. *T*_21_, *T*_22_ and *T*_23_ represented the relaxation time of bound water, immobilized water and free water, respectively.

**Figure 2 foods-12-01885-f002:**
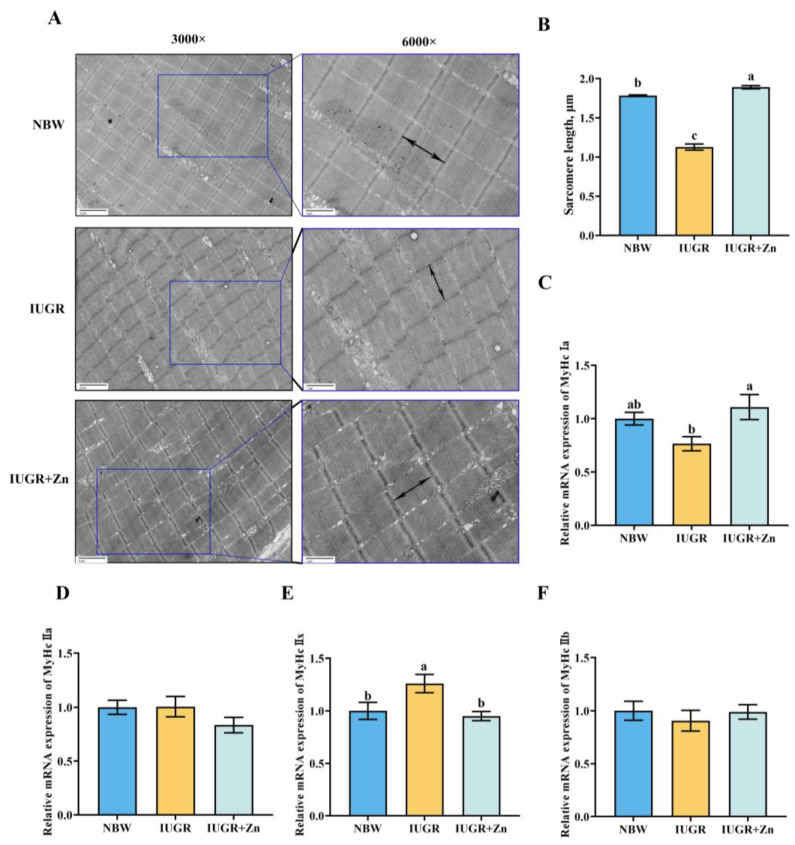
Effects of Nano-ZnO supplementation on the ultrastructure and the mRNA expression of muscle fiber characteristics of LD tissues in the IUGR pigs (*n* = 6). NBW, normal-born weight pigs; IUGR, intrauterine growth restriction pigs; IUGR + Zn, IUGR pigs fed with diets supplemented with 600 mg/kg Nano-ZnO. (**A**) The ultrastructure of LD tissues; (**B**) Sarcomere length of the LD tissues; (**C**–**F**) The mRNA expressions of *MyHc Ia*, *MyHc IIa*, *MyHc IIx*, *MyHc IIb*, respectively. Mean values with a common superscript do not differ.

**Figure 3 foods-12-01885-f003:**
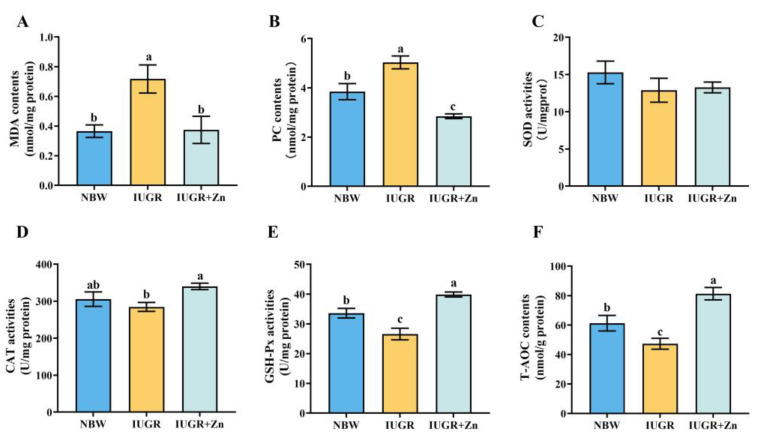
Effects of Nano-ZnO supplementation on the antioxidant status of LD tissues in the IUGR pigs (*n* = 6). NBW, normal-born weight pigs; IUGR, intrauterine growth restriction pigs; IUGR + Zn, IUGR pigs fed with diets supplemented with 600 mg/kg Nano-ZnO. (**A**) MDA, malondialdehyde; (**B**) protein carbonyls; (**C**) superoxide dismutase; (**D**) CAT, catalase; (**E**) GSH-Px, glutathione peroxidase; (**F**) T-AOC, total antioxidant capacity. Mean values with a common superscript do not differ.

**Figure 4 foods-12-01885-f004:**
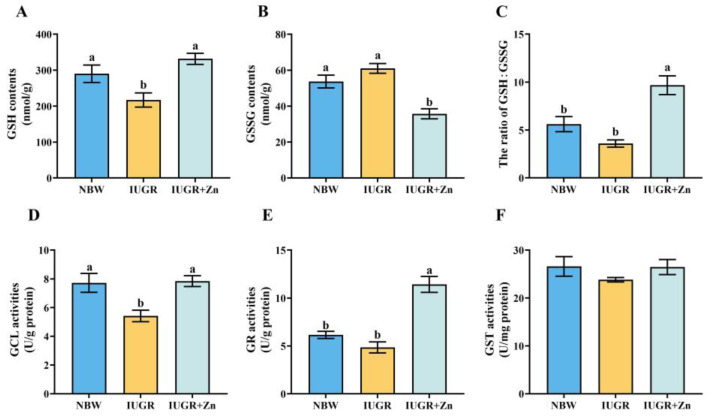
Effects of Nano-ZnO supplementation on the glutathione-related indicators of LD tissues in IUGR pigs (*n* = 6). NBW, normal-born weight pigs; IUGR, intrauterine growth restriction pigs; IUGR + Zn, IUGR pigs fed with diets supplemented with 600 mg/kg Nano-ZnO. (**A**) GSH, glutathione; (**B**) GSSG, oxidized glutathione; (**C**) the ratio of GSH/GSSG; (**D**) GCL, γ-glutamyl cysteine ligase; (**E**) GR, glutathione reductase; (**F**) GST, glutathione S-transferase. Mean values with a common superscript do not differ.

**Figure 5 foods-12-01885-f005:**
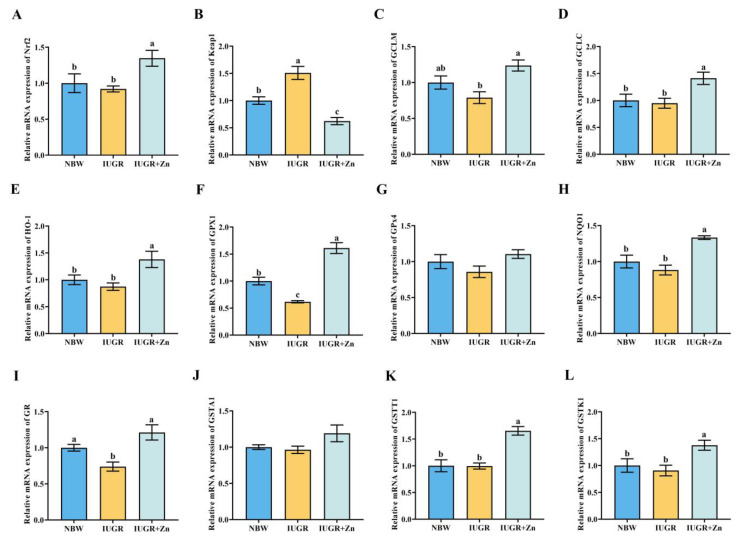
Effects of Nano-ZnO supplementation on the genes related to the Nrf2-GCL signaling pathway of LD tissues in the IUGR pigs (*n* = 6). NBW, normal-born weight pigs; IUGR, intrauterine growth restriction pigs; IUGR + Zn, IUGR pigs fed with diets supplemented with 600 mg/kg Nano-ZnO. (**A**) *Nrf2*, nuclear factor erythroid-derived 2-like 2; (**B**) *Keap1*, Kelch-like ECH-associated protein 1; (**C**) *GCLM*, glutamate-cysteine ligase modifier subunit; (**D**) *GCLC*, glutamate-cysteine ligase catalytic subunit; (**E**) *HO-1*, heme oxygenase 1; (**F**,**G**) *GPX*, glutathione peroxidase; (**H**) *NQO1*, NAD(P)H dehydrogenase, quinone 1; (**I**) *GR*, glutathione reductase; (**J**) *GSTA1*, glutathione S-transferase alpha 1; (**K**) *GSTT1*, glutathione S-transferase theta-1; (**L**) *GSTK1*, glutathione S-transferase kappa 1. Mean values with a common superscript do not differ.

**Figure 6 foods-12-01885-f006:**
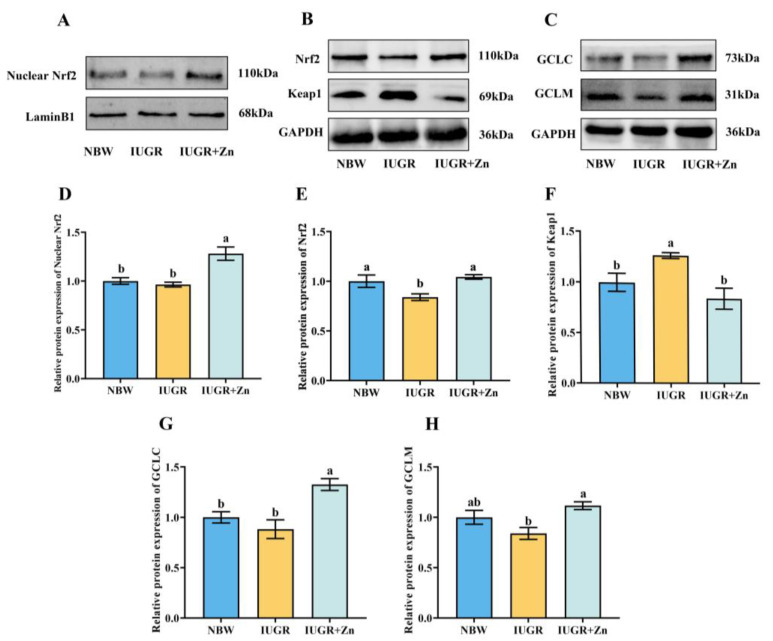
Effects of Nano-ZnO supplementation on the expression of proteins related to the Nrf2-GCL signaling pathway of LD tissues in IUGR pigs (*n* = 6). NBW, normal-born weight pigs; IUGR, intrauterine growth restriction pigs; IUGR + Zn, IUGR pigs fed with diets supplemented with 600 mg/kg Nano-ZnO. (**A**) Western blot of nuclear Nrf2 and LaminB1 proteins; (**B**) Western blot of Nrf2, Keap1 and GAPDH proteins; (**C**) Western blot of GCLC, GCLM and GAPDH proteins; (**D**) Relative expression of proteins nuclear Nrf2 /LaminB1; (**E**–**H**) Relative expression of proteins Nrf2/GAPDH, Keap1/GAPDH, GCLC/GAPDH, GCLM/GAPDH. Mean values with a common superscript do not differ.

**Figure 7 foods-12-01885-f007:**
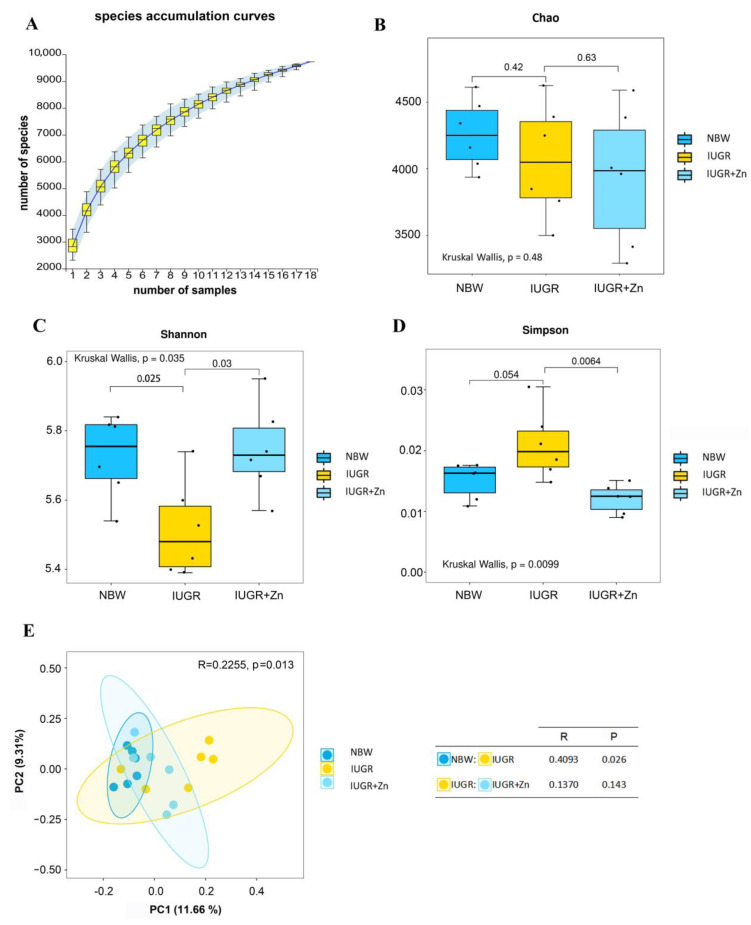
Effects of Nano-ZnO supplementation on the alpha and beta diversity of cecal microbiota in IUGR pigs (*n* = 6). NBW, normal-born weight pigs; IUGR, intrauterine growth restriction pigs; IUGR + Zn, IUGR pigs fed with diets supplemented with 600 mg/kg Nano-ZnO. (**A**) Species accumulation curves; (**B**) Chao index of OTU level; (**C**) Shannon index of OTU level; (**D**) Simpson index of OTU level; (**E**) Principal coordinates analysis (PCoA) of all samples and analysis of similarities (ANOSIM).

**Figure 8 foods-12-01885-f008:**
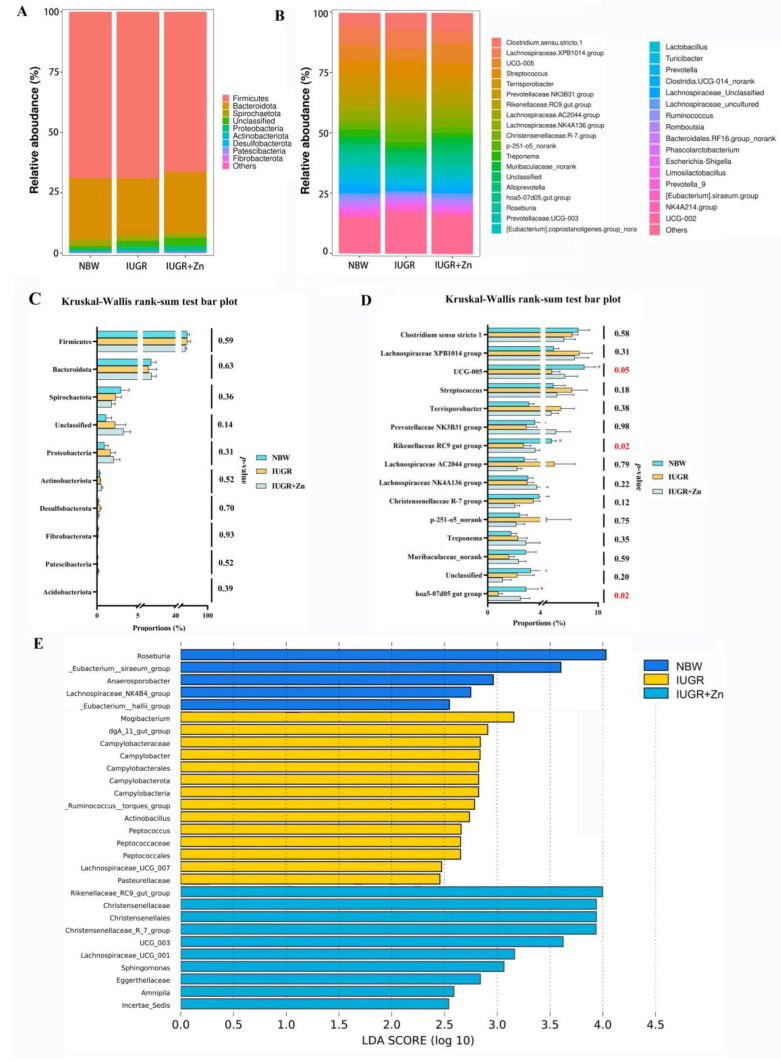
Effects of Nano-ZnO supplementation on the cecal microbiota compositions in the IUGR pigs (*n* = 6). NBW, normal birth weight pigs; IUGR, intrauterine growth restriction pigs; IUGR + Zn, IUGR pigs fed with diets supplemented with 600 mg/kg Nano-ZnO. (**A**) Relative abundance of the cecum microbiota at phylum level; (**B**) Relative abundance of the cecum microbiota at the genus level; (**C**) Composition difference at the phylum level; (**D**) Composition difference at the genus level, the “*” indicate statistical differences among groups (*p* < 0.05), and the *p*-values are marked in red; (**E**) The different bacterial taxa enriched in each group, and the linear discriminant analysis effect size (LEfSe) bar is determined by the discriminant analysis (LDA) scores (threshold ≥ 2 and *p* < 0.05) of the cecum samples.

**Figure 9 foods-12-01885-f009:**
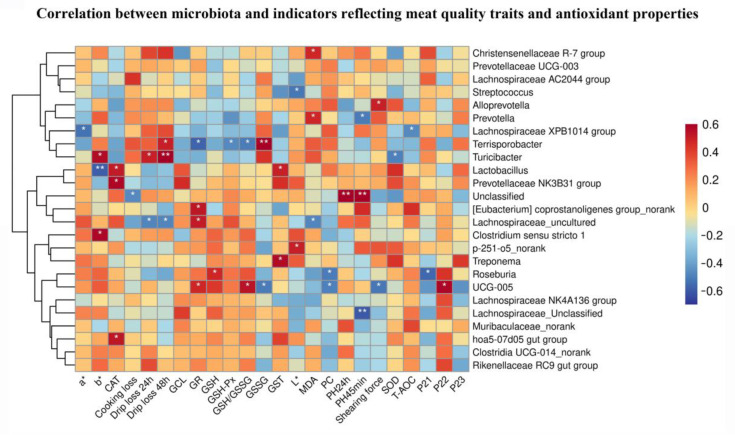
The correlation analysis between cecal microbiota and the indicators reflecting meat quality traits and antioxidant properties. The Pearson correlation coefficient was used in correlation analysis (*n* = 6). * *p* ≤ 0.05, ** *p* ≤ 0.01.

**Table 1 foods-12-01885-t001:** Effects of Nano-ZnO supplementation on meat quality of LD tissues in the IUGR pigs.

Item ^1^	Treatment ^2^			SEM	*p*
	NBW	IUGR	IUGR + Zn		
pH_45min_	6.47	6.55	6.78	0.07	0.200
pH_24h_	5.86	5.91	6.19	0.07	0.107
Drip loss_24h_ (%)	2.50	3.20	2.51	0.16	0.138
Drip loss_48h_ (%)	4.38 ^b^	6.07 ^a^	4.05 ^b^	0.28	0.002
Lightness *L**	43.41	43.99	43.43	0.90	0.961
Redness *a**	11.90 ^ab^	9.96 ^b^	14.08 ^a^	0.57	0.004
Yellowness *b**	20.17	20.09	20.17	0.37	0.995
Cooking loss (%)	29.9 ^ab^	34.52 ^a^	24.02 ^b^	1.59	0.015
Shearing force (N)	55.30 ^ab^	71.75 ^a^	38.52 ^b^	4.67	0.006

^1^ pH_45 min_, pH at 45 min post-mortem; pH_24h_, pH at 24 h post-mortem; Drip loss_24h_, drip loss at 24 h post-mortem; Drip loss_48h_, drip loss at 48 h post-mortem. ^2^ NBW, normal-born weight pigs; IUGR, intrauterine growth restriction pigs; IUGR + Zn, IUGR pigs fed with diets supplemented with 600 mg/kg Nano-ZnO. Data were shown as the mean values with pooled standard error (*n* = 6). Mean values with a common superscript do not differ.

**Table 2 foods-12-01885-t002:** Effects of Nano-ZnO supplementation on the peak area ratios (*P*_21_, *P*_22_ and *P*_23_) of water molecules of LD tissues in the IUGR pigs.

Item ^1^	Treatment ^2^			SEM	*p*
	NBW	IUGR	IUGR + Zn		
*P*_21_ (%)	4.25 ^b^	5.18 ^a^	4.94 ^a^	0.12	<0.001
*P*_22_ (%)	95.07 ^a^	94.07 ^b^	94.84 ^a^	0.09	0.021
*P*_23_ (%)	0.67	0.75	0.21	0.13	0.167

^1^ *P*_21_, peak area ratio of bound water; *P*_22_, peak area ratio of immobilized water; *P*_23_, peak area ratio of free water. ^2^ NBW, normal-born weight pigs; IUGR, intrauterine growth restriction pigs; IUGR + Zn, IUGR pigs fed with diets supplemented with 600 mg/kg Nano-ZnO. Data were shown as the mean values with pooled standard error (*n* = 6). Mean values with a common superscript do not differ.

## Data Availability

The data during this investigation are available from the corresponding author upon reasonable request.

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
