# Peer review of "Effects of Dietary Nano-Zinc Oxide Supplementation on Meat Quality, Antioxidant Capacity and Cecal Microbiota of Intrauterine Growth Retardation Finishing Pigs"

_foods, 2023, doi:10.3390/foods12091885_

Round 1

Reviewer 1 Report

line 84: IUGR can not be determined only by birth weight. PLease provide photo of them. Also give information about the litter size of the sows.

line 99: nutrient content

line 107: maybe better to say – rapidly frozen

line 140: what modifications? what was the setup of the machine?

line 236 and in all other tables and figures: If you have double superscript letter at any of the means than you are no longer looking for different superscripts but for similar ones. So the sentence you used logically incorrect. The correct one is: Mean values with a common superscript do not differ (p > 0.05)

Author Response

Response to Reviewer 1 Comments

Point 1: line 84: IUGR can not be determined only by birth weight. Please provide photo of them. Also give information about the litter size of the sows.

Response 1: Thanks very much for your valuable comments. In our study, newborn piglets that weighed within 0.5 standard deviation (SD) of the mean birth weight (BW) of the littermates were defined as normal birth weight (NBW), whereas those with 2 SD lower BW were defined as IUGR. The selection criteria for IUGR pigs used in our study referred to the methods commonly used in our laboratory and other published literature (Zhang et al., 2019; Zhang et al., 2023; Wu et al., 2006; D’Inca R et al., 2010). As you mentioned that IUGR can not be determined only by birth weight. But as it is easy to measure practically on farms and in clinics, fetal weight or birth weight relative to gestational age is often used as a criterion to detect IUGR, therefore, low birth weight is one of the most important characteristics of IUGR pigs. We are sorry to tell you that we didn’t take the photos of IUGR piglets at that time.

The liter size and parity number of sows are as follows.

Sow number

Litter size

Parity number

1

12

3

2

11

3

3

12

2

4

13

3

5

11

3

6

11

2

The corresponding references are listed as below:

Zhang, H.; Chen, Y.; Li, Y.; Zhang, T.; Ying, Z.; Su, W.; Zhang, L.; Wang T. L-Threonine improves intestinal mucin synthesis and immune function of intrauterine growth retarded weanling piglets. Nutrition. 2019, 59, 182-187.

 Zhang, X.; Yun, Yang.; Lai, Zheng.; Ji, S.; Yu, G.; Xie, Z.; Zhang, H.;Zhong, X.; Wang, T.; Zhang, L. Supplemental Clostridium butyricum modulates lipid metabolism by reshaping the gut microbiota composition and bile acid profile in IUGR suckling piglets. J Anim Sci Biotechnol. 2023, 14, 36.

Wu, G.; Bazer, FW.; Wallace, JM.; Spencer, TE. Board-invited review: intrauterine growth retardation: implications for the animal sciences. J Anim Sci. 2006. 84(9), 2316-2337.

D’Inca, R.; Kloareg, M.; Gras-Le Guen, C.; Le Huërou-Luron. Intrauterine growth restriction modifes the developmental pattern of intestinal structure, transcriptomic profle, and bacterial colonization in neonatal pigs. J Nutr. 2010, 140(5), 925-931.

Point 2: line 99: nutrient content

Response 2: Thanks very much for your valuable comments. We have changed the “nutritional contents” into “nutrient content”, please see line 99.

Point 3: line 107: maybe better to say – rapidly frozen

Response 3: Thanks for your valuable suggestions. We have changed “rapidly chilled” into “rapidly frozen”, please see line 107.

Point 4: line 140: what modifications? what was the setup of the machine?

Response 4: Thanks for your valuable suggestions. I’m sorry for the unclear and generalized expressions in the manuscript. The modifications were as follows, in our study, the LD samples were left left to cool to room temperature under running water after being cooked in a water bath at 80 ℃, then cut into uniform cubes of 1×1×1.5 cm3, and the shearing force were measured by Digital Meat Tenderness Meter (C-LM3B, Northeast Agricultural University, Harbin, China). However, the cited literature indicated that the samples were cooked at 75 ℃, cut into 1×1×1 cm3, and were measured by Texture Analyzer (TMS-Touch, Food Technology Corp., USA.) The parameters of the machine in our study were as follows: pretest speed, 0.5 mm/s; test speed, 1.0 mm/s; trigger type, auto-5 g; and strain, 50%.

Point 5: line 236 and in all other tables and figures: If you have double superscript letter at any of the means than you are no longer looking for different superscripts but for similar ones. So the sentence you used logically incorrect. The correct one is: Mean values with a common superscript do not differ

Response 5: Thanks for your valuable suggestions. We have changed “Mean values with different superscript letters were significantly different” into “Mean values with a common superscript do not differ” in all captions.

Reviewer 2 Report

- This manuscript has an alternative approach to ameliorating the impacts of intrauterine growth retardation on meat quality by using nano-zinc oxide. The authors can address and make the changes follow my comments below:

- L83-85: Please provide more details of the method for NBW and IUGR classification and selection rather than citing previous publications and pushing the readers to find by themselves.

- L91-92: The dose of 600 mg/kg was just tested in weaned pigs, and then authors assume it may be suitable. However, it is questionable and should be needed to investigate. How could the authors speculate without any research evidence?

- What were the ways that authors supplemented Nano-ZnO in three phases of the experiment?

- The feed intake probably was recorded, and the authors should provide a table of growth performance that includes feed intake. This will illustrate how much feed and Nano-ZnO the pigs ate which then may have an impact on meat quality.

- If possible, authors should analyze the results as a 2x2 factorial design (BW: NBW vs IUGR, and Nano-ZnO: control vs supplemented) to understand clearly the effect of BW on meat quality and if Nano-ZnO can ameliorate this impact.

Author Response

Response to Reviewer 2 Comments

Point 1: L83-85: Please provide more details of the method for NBW and IUGR classification and selection rather than citing previous publications and pushing the readers to find by themselves.

Response 1: Thanks for your valuable suggestions. We have added the details of the method for NBW and IUGR classification and selection. The added description of this point is “Normal birth weight (NBW) refers to newborn piglets who weigh within 0.5 standard deviations (SD) of their littermates’ mean birth weights (BW), while IUGR refers to newborns whose BW is 2 SD lower”, please see line 82-85.

Point 2: L91-92: The dose of 600 mg/kg was just tested in weaned pigs, and then authors assume it may be suitable. However, it is questionable and should be needed to investigate. How could the authors speculate without any research evidence?

Response 2: Thanks for your valuable suggestions. First of all, the dose of 600 mg/kg Nano-ZnO was based on our previous study, which manifested that the dose of 800 mg/kg Nano-ZnO (600 mg/kg Zn from Nano-ZnO, in line with the present study) showed better effects. Furthermore, our present experiment aims to investigate whether Nano-ZnO has a corresponding regulatory effect, and we will do further tests to clarify the appropriate dose at a later stage. We also referred to other animal tests, such as 5000 mg/kg Nano-ZnO in mice fed for 32 weeks without significant side effects. Although the super high dose of 5000mg/kg has some side effects on zinc metabolism, we found the same dose of nano-ZnO is safer compared to ZnSO4, and pigs tolerate zinc much better than mice, so we presume that the addition of only 600mg/kg of zinc from nano-ZnO, which has potential benefits, will not cause significant side effects. If you still have any question or suggestion, please give us one more message and we will reply you as soon as possible.

The corresponding references are listed as below:

Wang, C.; Zhang, L.; Ying, Z.; He, J.; Zhou, L.; Zhang, L.; Zhong, X.; Wang, T. Effects of dietary zinc oxide nanoparticles on growth, diarrhea, mineral deposition, intestinal morphology, and barrier of weaned piglets. Biol Trace Elem Res. 2018, 185, 364-374.

Wang, C.; Lu, J.; Zhou, L.; Li J.; Xu, J.; Li, W.; Zhang, L.; Zhong X.; Wang, T. Effects of Long-Term Exposure to Zinc Oxide Nanoparticles on Development, Zinc Metabolism and Biodistribution of Minerals in Mice. PLoS ONE. 2016, 11(10): e0164434.

Point 3: What were the ways that authors supplemented Nano-ZnO in three phases of the experiment?

Response 3: Thanks for your valuable suggestions. In our present study, for the Nano-ZnO treatment group (IUGR+Zn), the additional 600 mg/kg Nano-ZnO was directly added to the basal diet in three phases of the experiment. If you still have any question or suggestion, please give us one more message and we will reply you as soon as possible.

Point 4: The feed intake probably was recorded, and the authors should provide a table of growth performance that includes feed intake. This will illustrate how much feed and Nano-ZnO the pigs ate which then may have an impact on meat quality.

Response 4: Thanks for your valuable suggestions. As you mentioned, it is more appropriate to add the growth performance data in our manuscript to illustrate the intake of Nano-ZnO of pigs. However, we are regretful to tell you that the growth performance data has been published in another article previously (doi: 10.3389/fvets.2022.884945). The growth performance data are as follows.

Effects of dietary Nano-ZnO supplementation on growth performance of IUGR finishing pigs.

Items 1

Group 2

SEM

p -value

NBW

IUGR

IUGR+Zn

BW (Kg)

1 d

1.52a

0.96b

0.96b

0.06

<0.001

21 d

6.06a

4.88b

4.86b

0.16

<0.001

77 d

30.93a

27.53ab

26.16b

0.83

0.042

163 d

121.83a

113.63ab

108.54b

1.82

0.003

ADG (Kg / d)

1–21 d

0.22a

0.19b

0.19b

0.01

0.018

21–77 d

0.44

0.40

0.38

0.01

0.130

77–163 d

1.06a

1.00ab

0.96b

0.02

0.028

ADFI (Kg / d)

21–77 d

0.87

0.78

0.72

0.03

0.114

77–163 d

2.53a

2.44a

2.16b

0.05

0.002

G: F

21–77 d

0.52

0.52

0.55

0.03

0.746

77–163 d

0.42

0.41

0.45

0.01

0.220

1 BW, body weight; ADG, average daily gain; ADFI, average daily feed intake; G: F, gain-to-feed ratio. Data were expressed as mean and SEM, n = 6;

a, b Means that values within a row with different superscript letters were significantly different (p < 0.05).

2NBW, normal birth weight pigs; IUGR, intrauterine growth restriction pigs; IUGR+Zn, IUGR pigs fed with diets supplemented with 600 mg Zn/kg from Nano-ZnO.

Point 5: If possible, authors should analyze the results as a 2x2 factorial design (BW: NBW vs IUGR, and Nano-ZnO: control vs supplemented) to understand clearly the effect of BW on meat quality and if Nano-ZnO can ameliorate this impact.

Response 5: Thanks very much for your valuable comments. It is necessary to set up the NBW+Zn group to carried out the results as a 2×2 factorial design, so that we can clearly understand the impacts of both BW and Nano-ZnO on the meat quality of pigs. However, the purpose of this study was to investigated the regulated effect of Nano-ZnO on IUGR pigs, and the NBW, IUGR and IUGR+Zn group were carried out in the current study. Before this study stared, we had also combined the common experiment design in our lab and other published literature (Zhang et al., 2023). But the 2 × 2 factorial design will surely improve the rigor and integrity of the study. We will pay attention to this and try to add another treatment group in future studies. Thanks very much for your valuable suggestions again. If you still have any question or suggestion, please give us one more message and we will reply you as soon as possible.

Reviewer 3 Report

Dear authors, interesting manuscript, however, I have some concerns regarding the introduction, aim, discussion and conclusions. Please see the attached file

Best

Author Response

Response to Reviewer 3 Comments

Point 1: Line 61: here I suggest to cite: 10.3390/ani10122386 and 10.3390/ani10060945

Response 1: Thanks for your valuable suggestions. We have cited these 2 articles according to your valuable suggestion. Please see line 60.

Point 2: Line 72-75: please rewrite the aim of the experimet, it is not clear

Response 2: Thanks for your valuable suggestions. We have seriously considered your suggestions, and deleted the redundant expressions and rewritten the aim of this study to make it clearer. Please see Line 71-74.

Point 3: Line 82: report the number per parity and state why it is comparable

Response 3: Thanks for your valuable suggestions. All sows had litters in the range of 10-12, and all in their second or third litters.

The liter size and parity number of sows are as follows.

Sow number

Litter size

Parity number

1

12

3

2

11

3

3

12

2

4

13

3

5

11

3

6

11

2

Point 4: Line 131: explain why “modified”? 

Response 4: Thanks for your valuable suggestions. I’m sorry for the unclear and generalized expressions in the manuscript. Most of the steps in this study to determine the cooking loss were similar to those in the cited literature, with the minor difference that the samples were cooled to room temperature under running water after cooking in the manuscript, while the cooling method in the cited literature was natural cooling. If you still have any question or suggestion, please give us one more message and we will reply you as soon as possible.

Point 5: Line 218: did you perform distribution analysis?

Response 5: Thanks for your valuable suggestions. We reviewed relevant information before conducting the data analysis. Although ANOVA theoretically requires that the overall data satisfy a normal distribution, the normality test has more stringent judgment criteria, and in practice, we chose to conduct ANOVA directly after the homogeneity test of variancetest. Previous experience also shows that the results obtained in this statistical way are often credible and robust. We will pay attention to this and try to perform distribution analysis in future studies. Thanks very much for your valuable suggestions again.

Point 6: Line 374: figures are not clear, too small

Response 6: Thanks for your valuable suggestions. We have adjusted the size of Fig.8 to make it clearer for the readers.

Point 7: Line 392: usually the discussion start with repeating the aim

Response 7: Thanks for your valuable suggestions. We have repeated the aim of where the discuss started. Please see line 386-389.

Point 8: Line 394: here cite: 10.3390/ani10060945 and 10.3390/ani10122386

Response 8: Thanks for your valuable suggestions. We have cited these 2 articles according to your valuable suggestion. Please see line 391.

Point 9: Line 517: please report practical implications and limitations of the study

Response 9: Thanks for your valuable suggestions. We have seriously considered your suggestions, and we have added the practical implications of the present study, please see line 509-511. Furthermore, we have adjusted the expression of the study according to your valuable suggestions, please see line 512-518.

Point 10: Line 518: please make the conclusions more clear

Response 10: Thanks for your valuable suggestions. We have deleted the redundant and unclear expression of conclusions to make it concise and accurate. The specific modifications are as follows. (1) We have corrected the confusing sentences: The “Significant changes in the cecal microbial composition of IUGR pigs treated with Nano-ZnO were manifested by changes in alpha diversity and the increased abundance of UCG-005 and hoa5-07d05 gut group, with which the improved meat quality and enhanced antioxidant capacity may be closely correlated.” has been changed into “Nano-ZnO supplementation changed the alpha diversity and increased the UCG-005 and hoa5-07d05 gut group abundance of cecal microbiota, which may closely correlate with the improved meat quality and enhanced antioxidant capacity.” (2) We have deleted the redundant sentences “The present investigation may provide a new idea to study the regulatory effect of Nano-ZnO-mediated gut microbiota on the development and metabolism of muscle”. Please see line 524-527. If you still have any question or suggestion, please give us one more message and we will reply you as soon as possible. Thanks very much for your valuable suggestions again.

Round 2

Reviewer 3 Report

The paper after the revisions improved a lot